# Are we advancing universal health coverage through cataract services? Protocol for a scoping review

Chan Ning Lee,[1,2] Jacqueline Ramke [ID] ,[1,3] Ian McCormick,[1] Justine H Zhang [ID] ,[1,4] Ada Aghaji,[1,5] Nyawira Mwangi [ID] ,[1,6] Helen Burn,[1,7] Iris Gordon,[1] Mayinuer Yusufu,[8] Mingguang He,[9,10] Juan Carlos Silva,[11] Matthew J Burton[1,12]

For numbered affiliations see end of article.

**Correspondence to**
Dr Jacqueline Ramke;
jacqueline.ramke@lshtm.ac.uk

## ABSTRACT

**Introduction** Universal health coverage (UHC) includes the dimensions of equity in access, quality services that improve health and protection against financial hardship. Cataract continues to be the leading cause of blindness globally, despite cataract surgery being an efficacious intervention. The aim of this scoping review is to map the nature, extent and global distribution of data on cataract services for UHC in terms of equity, access, quality and financial protection.

**Methods and analysis** The search will be constructed by an Information Specialist and undertaken in MEDLINE, Embase and Global Health databases. We will include all published non-interventional primary research studies and systematic reviews that report a quantitative assessment of access, equity, quality or financial protection of cataract surgical services for adults at the subnational, national, regional or global level from population-based surveys or routinely collected health service data since 1 January 2000 and published through to February 2020. Screening and data charting will be undertaken using Covidence systematic review software. Titles and abstracts of identified studies will be screened by two authors independently. Full-text articles of potentially relevant studies will be obtained and reviewed independently by two authors against the inclusion criteria. Any discrepancies between the authors will be resolved by discussion, and with a third author as necessary. A data charting form will be developed and piloted on three studies by three authors and amendments made as necessary. Data will be extracted by two reviewers independently and summarised narratively and using maps.

**Ethics and dissemination** Ethical approval was not sought as the scoping review will only use published and publicly accessible data. The review will be published in an open access peer-reviewed journal. A summary of the results will be developed for website posting, stakeholder meetings and inclusion in the ongoing *Lancet Global Health* Commission on Global Eye Health.

### Strengths and limitations of this study

► The broad scope of this review will result in the first synthesis to date of data on the universal health coverage dimensions of cataract surgical services.

► Another strength is that we will include studies from all world regions and high-income, low-income and middle-income countries with no language restrictions, to give a global picture of cataract services.

► A potential limitation is the paucity of available information on the 'financial protection' dimension.

provision of quality eye care services contributes directly to achieving universal health coverage (UHC).[1] WHO estimated that in 2020 up to 2.2 billion people have some form of vision impairment and that this figure is projected to rise leading to an increased burden on health systems.[1]

### Monitoring progress towards UHC

The WHO defines UHC in the following terms:

> Universal health coverage (UHC) means that all people and communities can use the promotive, preventive, curative, rehabilitative and palliative health services they need, of sufficient quality to be effective, while also ensuring that the use of these services does not expose the user to financial hardship.[2]

It has three broad principles: (1) equity in access; (2) quality services that improve health; (3) protection against financial risk. WHO and the World Bank have developed a framework for tracking progress towards UHC.[3] This focuses on two elements:

1. Measuring the coverage of essential health services, as a proportion of the population that can access essential quality health services.

## INTRODUCTION

Eye health and vision impairment represent a major global health concern. In the recent *World Report on Vision,* WHO outlined how the

2. Measuring financial protection by determining the proportion of the population in whom direct payment made to obtain health services leads to financial hardship and/or a threat to living standards.[3]

A variety of different types of indicators are used. Effective service coverage indicators measure the proportion of people in need of services who receive services of sufficient quality to obtain potential health gains; these are preferred if available. Service coverage indicators measure the proportion of the population that needs a service that receive it. Proxy indicators are sometimes used where service coverage indicators are not available, but provide a correlated indication of the provision of a health service.[3 4]

The WHO and World Bank have selected a panel of 16 'tracer indicators' to monitor progress towards UHC.[2] There is currently no eye health related indicator among this panel, though effective cataract surgical coverage (eCSC) and effective refractive error coverage were proposed in WHO's Thirteenth General Programme of Work 2019–2023 Impact Framework.[5] In addition to measuring population-level coverage, it is important to measure equity in service provision, by disaggregating the data and comparing subpopulations such as wealth quintiles, education, sex, age and geographical region.[3]

### Monitoring cataract services for UHC
Cataract is the leading cause of blindness globally and is the second leading cause of vision impairment.[6] The last three decades have seen a marked increase in available data on vision impairment due to cataract, as well as cataract services. These data have enabled calculation of indicators of access and quality of cataract surgery, including:

► Cataract surgical rate (CSR): the number of cataract operations per million population per year.[7]
► CSC: the number of people in a population who have received cataract surgery as a proportion of those having operable and operated cataract.[6]
► Cataract surgical outcome: the presenting and/or best-corrected visual acuity of the operated eye.
► ECSC: the number of people in a population with operated cataract and a visual acuity of 6/18 or better as a proportion of those having operable and operated cataract.[8]

eCSC, has the characteristics of an effective service coverage indicator, as preferred by the WHO/World Bank, as it combines information on the proportion of the population covered and the outcome of the surgical intervention.[9 10]

Disaggregation of larger datasets has allowed analyses of equity in cataract surgery as well, for example, to highlight existing gender disparities in CSC.[11–15] Much less data are available quantifying financial aspects of cataract services. To our knowledge, no existing synthesis of the distribution and quantity of known evidence for the UHC dimensions of cataract surgery has been undertaken. The aim of this scoping review is to map the nature, extent and global distribution of data on cataract surgical services

for UHC in terms of equity, access, quality and financial protection.

We chose to undertake a scoping review rather than an alternative evidence synthesis approach because we wished to identify and map the available evidence, which we anticipate will be heterogeneous.[16]

## METHODS
### Objectives/scoping review questions
We aim to answer the following two questions in relation to cataract services for UHC:

1. What is the nature, extent and global distribution of data on the coverage and effectiveness of cataract services?
2. What is the nature, extent and global distribution of data on financial protection in relation to cataract services?

### Protocol and registration
The protocol for this scoping review is reported according to the relevant sections of the Preferred Reporting Items for Systematic Reviews and Meta-Analyses Extension for Scoping Reviews (PRISMA) guideline (online supplementary annex 1).[17] The protocol is registered on the Open Science Framework (https://osf.io/k3mwg/).

### Eligibility criteria
We will include all published prospective and retrospective primary research studies and systematic reviews that report a quantitative assessment of access, equity, quality or financial protection of cataract surgical services for adults at the subnational, national, regional or global level (examples outlined in table 1). We will include population-level observational studies and reports, including those that use routinely collected data (such as in health information systems) and household surveys such as Rapid Assessment of Avoidable Blindness (RAAB) surveys. We will exclude intervention studies and studies within clinical subpopulations as their outcomes can be different to the general population (eg, people with diabetes, people with age-related macular degeneration). We will exclude studies focused exclusively on cataract services for children (aged under 18 years), as these services differ substantially from those for age-related cataract.

To assess access we will include studies that report CSC and CSR, which are priority indicators for monitoring global eye health.[18] Beyond these, we will include studies that report the number and distribution of human resources and surgical facilities. We acknowledge that there are many quantitative and qualitative elements of healthcare quality as defined by WHO.[19] However, for the purposes of this review, we will focus on only three— eCSC, vision outcomes of cataract surgery and reported complications. We anticipate the literature reporting financial protection associated with cataract surgery will be limited. We will include studies and surveys that report

**Table 1** Primary and secondary outcomes included in the review, mapped against UHC dimensions

| UHC dimension | Primary cataract indicator | Secondary cataract indicator |
|---|---|---|
| Access (Coverage)—the availability of good health services within reasonable reach and available at the point of need. | Cataract surgical coverage | ► Cataract surgical rate[23]<br>► No of operating surgeons by country<br>► No and distribution of operating centres by country |
| Quality—limited to the WHO quality elements of effectiveness and safety | Effective cataract surgical coverage[8] | ► Cataract surgical outcome[11]<br>► Complication rates per surgeon/institution |
| Financial Protection—direct payments made to obtain health services do not expose people to financial hardship and do not threaten living standards | Rate of Catastrophic Spending on cataract surgery (25% of total household expenditure per WHO)<br><br>Rate of Impoverishing Spending on cataract surgery (relative to national or international poverty line) | Cost of cataract Surgery (to patient/household)[24 25] |
| Equity—services are accessible to all who need them | Disaggregation of any of the primary or secondary indicators by sex/gender[13 15 26] | Disaggregation of any of the primary or secondary indicators by any other PROGRESS factor[20]:<br><br>Place of residence (eg, urban/rural, subnational unit)<br><br>Race/ethnicity/culture/language<br><br>Occupation<br><br>(Gender/sex)<br><br>Religion<br><br>Education<br><br>Socioeconomic status[12]<br><br>Social capital (eg, marital status)[27] |

'catastrophic' and 'impoverishing' spending on cataract surgery according to WHO definitions, as well as other related measures of personal and government expenditure on cataract surgery and service provision (table 1). We will use the PROGRESS acronym[20] to assess equity: Place of residence; Race/ethnicity/culture/language; Occupation; Gender/sex; Religion; Education; Socioeconomic status; Social capital/ networks.

Studies will be limited to those including data collected since 1 January 2000 to provide a contemporary view of cataract services. The search strategy will be undertaken without language restrictions and translation will be arranged when required.

### Search strategy

We will search Embase, MEDLINE and Global Health databases for studies published from 1 January 2000 to February 2020 using search strategies developed by an Information Specialist from Cochrane Eyes and Vision (IG) (MEDLINE Search Strategy included in online supplementary annex 2). We will provide a list of included studies and reports to field experts and request they identify additional sources of both published and unpublished reports for consideration in the review.

We will use the RAAB repository (http://raabdata.info/) to identify all reports and data from sub-national and national RAAB studies taken from 1 January 2000 onwards.

To identify government and non-government reports in the grey literature, we will use a checklist adapted from the Canadian Agency for Drugs and Technologies in Health Grey Matters checklist to undertake a search of relevant websites.[21]

The following grey literature databases and repositories will be searched:
► OpenGrey (http://www.opengrey.eu/).
► Global Burden of Disease (http://www.healthdata.org/gbd).
► Global Health Data Exchange (http://ghdx.healthdata.org/).
► WHO (https://www.who.int/library/en/).
► International Agency for the Prevention of Blindness (https://www.iapb.org/global-vision-database-maps/).
► National Ministry of Health websites.

### Selection of sources of evidence

All titles and abstracts will be screened by at least two investigators independently using Covidence systematic review software (Veritas Health Innovation, Melbourne, Australia; available at www.covidence.org). Assessment of eligibility for inclusion will be carried out by two

investigators independently with a third investigator reviewing discrepancies. Reference lists of all included articles will be examined to identify further potentially relevant reports. The study selection process will be summarised in a PRISMA flow diagram.

## Data charting

The data charting form will be developed in Covidence and piloted by investigators prior to use. Data charting will be carried out by two investigators independently. As data sources are expected to be heterogeneous and broad in nature, data charting will be an iterative process throughout the review. Information that is absent or unclear will be addressed by contacting study authors with up to three attempts by email. The result of these attempts will be reported.

Because our focus is on mapping the availability of evidence, we did not to undertake quality appraisal of individual studies.[17]

## Data items
### Source characteristics
► Published Data Characteristics—Author(s), Year of Publication, Journal, Language.
► Grey Literature Characteristics—Author (Organisation, eg, WHO, Ministry of Health), Year of Publication, Source Website (eg, government/non-government organisation), Language, Type of Literature (Report, Thesis, Technical Report, Statistics, other).

### Study characteristics
Type of study, countries/regions investigated, level of analysis (subnational, national, regional, global), sample details (frame, size), year of data collection, outcome(s) reported (as outlined in table 1), UHC dimension(s) investigated (Access, Equity, Quality, Financial Protection).

## Synthesis of results

Following data charting, we will undertake narrative synthesis. Where possible maps will be used to summarise the available global, regional, national and subnational distribution and proportion of studies reporting each UHC dimension for cataract surgery. Tables will be constructed to demonstrate distribution of studies by region and (if appropriate) country. Where enough data are identified, further quantitative analyses of primary or secondary outcomes may be undertaken as a subsequent analysis.

## Patient and public involvement

This protocol was developed with input from the Commissioners of the *Lancet Global Health* Commission on Global Eye Health,[22] which includes people with lived experience of vision impairment (and cataract surgery), policymakers, academics, clinicians, government eye health programme leaders and advocacy specialists.

## ETHICS AND DISSEMINATION

As this scoping review will only consider publicly available literature and reports, no ethics approval was sought.

Findings will be published in an open-access peer-reviewed journal, and a summary will be developed for website access and stakeholder meetings. A summary of the findings will also be included in the ongoing *Lancet Global Health* Commission on Global Eye Health.[22]

**Author affiliations**
[1]International Centre for Eye Health, London School of Hygiene and Tropical Medicine, London, United Kingdom
[2]St Paul's Eye Unit, Royal Liverpool University Hospital, Liverpool, United Kingdom
[3]School of Optometry and Vision Science, University of Auckland, Auckland, New Zealand
[4]Manchester Royal Eye Hospital, Manchester, United Kingdom
[5]Department of Opthalmology, University of Nigeria, Enugu, Nigeria
[6]Department of Clinical Medicine, Kenya Medical Training College, Nairobi, Kenya
[7]Department of Ophthalmology, Stoke Mandeville Hospital, Aylesbury, United Kingdom
[8]Beijing Institute of Ophthalmology, Beijing Tongren Eye Center, Beijing Tongren Hospital, Capital Medical University, Beijing, China
[9]State Key Laboratory of Ophthalmology, Zhongshan Ophthalmic Center, Sun Yat-sen University, Guangzhou, Guangdong, China
[10]Centre for Eye Research Australia, Royal Victorian Eye and Ear Hospital, University of Melbourne, Melbourne, Victoria, Australia
[11]Division of Blindness Prevention, Pan American Health Organization, Bogota, Colombia
[12]Moorfields Eye Hospital, London, United Kingdom

**Contributors** JR and MJB conceived the idea for the review. CNL, JR and MJB drafted and revised the protocol with suggestions from IM, JHZ, AA, NM, HB, IG, MY, MH and JCS. IG constructed the search.

**Funding** MJB is supported by the Wellcome Trust (207472/Z/17/Z). JR is a Commonwealth Rutherford Fellow, funded by the UK government through the Commonwealth Scholarship Commission in the UK. The Lancet Global Health Commission on Global Eye Health is supported by The Queen Elizabeth Diamond Jubilee Trust, Moorfields Eye Charity (grant number GR001061), NIHR Moorfields Biomedical Research Centre, Wellcome Trust, Sightsavers, The Fred Hollows Foundation, The SEVA Foundation, British Council for the Prevention of Blindness and Christian Blind Mission.

**Competing interests** None declared.

**Patient and public involvement** Patients and/or the public were not involved in the design, or conduct, or reporting, or dissemination plans of this research.

**Patient consent for publication** Not required.

**Provenance and peer review** Not commissioned; externally peer reviewed.

**ORCID iDs**
Jacqueline Ramke http://orcid.org/0000-0002-5764-1306
Justine H Zhang http://orcid.org/0000-0001-8385-2003
Nyawira Mwangi http://orcid.org/0000-0002-8236-470X

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
