## [Reviewer comments · BMJ Open]

ARTICLE DETAILS

TITLE (PROVISIONAL)	Are we advancing Universal Health Coverage through cataract services? Protocol for a scoping review
AUTHORS	Lee, Chan; Ramke, Jacqueline; McCormick, Ian; Zhang, Justine; Aghaji, Ada; Mwangi, Nyawira; Burn, Helen; Gordon, Iris; Yusufu, Mayinuer; He, Mingguang; Silva, Juan Carlos; Burton, Matthew J

VERSION 1 – REVIEW

REVIEWER	KM Saif-Ur-Rahman Health Systems and Population Studies Division, icddr,b, Bangladesh
REVIEW RETURNED	01-May-2020

GENERAL COMMENTS	This is a very good attempt. I have got a couple of methodological concerns which should be addressed to improve the quality of the scoping review and the protocol. Please find my comments here: 1. Abstract - In methods and analysis, there is nothing about quality appraisal.2. Is figure 1 necessary? That is well-known to the readers. Rather a conceptual framework of the proposed topics in the context of UHC should be there.3. Data charting and extraction: Should be conducted independently by two reviewers. Please mention that.4. There is nothing about quality appraisal of included articles. As you are going to include systematic reviews and primary studies, the quality assessment should be done using specific tools for assessing quality of different types of studies.
---

REVIEWER	Samuel Bert Boadi-Kusi University of Cape Coast, Ghana
REVIEW RETURNED	15-May-2020

GENERAL COMMENTS	I have carefully read through the methods of the study “Are we advancing the Universal Health Coverage through Cataract Services? Protocol for a scoping review” The aims of the review is to map the nature, extent and global distribution of data on cataract services for UHC in terms of equity, access, quality and financial protection. The methods are scientifically and it is my view that they can be replicated at any given time. However, on page 6, under Eligibility criteria, line 12: it will be ideal to indicate the state the two vision outcomes which the study will focus on although they are found in the tables.
--

	Again, on Page 7, Line 5: It will be ideal to state the cut off point for the data. It is good the stating point has been stated as 1 January 2000 but the cut off is equally relevant. I congratulate the authors for a good work.
--	---

VERSION 1 – AUTHOR RESPONSE

Reviewer 1

This is a very good attempt. I have got a couple of methodological concerns which should be addressed to improve the quality of the scoping review and the protocol. Please find my comments here:

1. Abstract - In methods and analysis, there is nothing about quality appraisal.

As outlined in the PRISMA-Scr guideline (Tricco et al <https://doi.org/10.7326/M18-0850>), quality assessment is considered optional in scoping reviews, depending on the aims of the review. Because our focus is on mapping the availability of evidence rather than quantifying it, we have opted not to undertake quality appraisal of individual studies. We believe this is in keeping with the best-practice guidance for scoping reviews.

2. Is figure 1 necessary? That is well-known to the readers. Rather a conceptual framework of the proposed topics in the context of UHC should be there.

We have removed Figure 1. We have retained the explanation of UHC and have mapped the relevant eye care outcomes for each dimension in Table 1. We believe this combination serves as a useful framework for readers and for the review process.

3. Data charting and extraction: Should be conducted independently by two reviewers. Please mention that.

The following was added to the Data Charting section:

Data charting will be carried out by two investigators independently.

4. There is nothing about quality appraisal of included articles. As you are going to include systematic reviews and primary studies, the quality assessment should be done using specific tools for assessing quality of different types of studies.

As outlined in response to your first comment, we decided not to undertake quality appraisal of individual studies, which is in keeping with the guidance for scoping reviews.

Reviewer 2

I have carefully read through the methods of the study “Are we advancing the Universal Health Coverage through Cataract Services? Protocol for a scoping review” The aims of the review is to map the nature, extent and global distribution of data on cataract services for UHC in terms of equity, access, quality and financial protection.

1. The methods are scientifically and it is my view that they can be replicated at any given time. However, on page 6, under Eligibility criteria, line 12: it will be ideal to indicate the state the two vision outcomes which the study will focus on although they are found in the tables.

We thought the table provided an overview of the dimensions, but we have now added text to this paragraph under Eligibility Criteria to briefly outline the outcomes for each of the UHC dimensions. For example:

To assess access we will include studies that report cataract surgical coverage and cataract surgical rate, which are priority indicators for monitoring global eye health.¹⁹ Beyond these, we will include studies that report the number and distribution of human resources and surgical facilities.

We will use the PROGRESS acronym²⁴ to assess equity.

2. Again, on Page 7, Line 5: It will be ideal to state the cut off point for the data. It is good the stating point has been stated as 1 January 2000 but the cut off is equally relevant.

We have added details in the Search Strategy section:

We will search Embase, MEDLINE and Global Health databases for studies published from 1 January 2000 through to February 2020 using search strategies developed by an Information Specialist from Cochrane Eyes and Vision.

We did not put the exact search date because this is the protocol, but we will put it in the manuscript that reports our results.

VERSION 2 – REVIEW

REVIEWER	K M Saif-Ur-Rahman Health Systems and Population Studies Division, icddr,b; Bangladesh
REVIEW RETURNED	29-May-2020

GENERAL COMMENTS	Thanks to the authors for explanations. As they have mentioned in the reply to the reviewer's comment, "As outlined in the PRISMA-Scr guideline (Tricco et al https://doi.org/10.7326/M18-0850), quality assessment is considered optional in scoping reviews, depending on the aims of the review. Because our focus is on mapping the availability of evidence rather than quantifying it, we have opted not to undertake quality appraisal of individual studies. We believe this is in keeping with the best-practice guidance for scoping reviews" - in my opinion, this should be added in the
--

	manuscript so that the readers will get an idea of why the risk of bias has not been assessed.
--	--

REVIEWER	Samuel Bert Boadi-Kusi University of Cape Coast, Ghana
REVIEW RETURNED	31-May-2020

GENERAL COMMENTS	The authors have attempted to revise the manuscript to the best of their abilities based on our earlier comments. There are however, minor comments which needs to be addressed. Abstract, line 3: the authors use " quality or financial protection" elsewhere the two are treated differently. Reading the manuscript, they are two different items so I suggest they replace "or" with "and" . This has been repeated on line 2 under " Eligibility" Data Charting and Extraction The authors suggests that information that is absent or unclear will be addressed by contacting the study authors. It is however unclear what the authors will do to such data if they do not receive any feedback
--

VERSION 2 – AUTHOR RESPONSE

Reviewer(s)' Comments to Author:

Reviewer: 1

Thanks to the authors for explanations. As they have mentioned in the reply to the reviewer's comment, "As outlined in the PRISMA-Scr guideline (Tricco et al <https://doi.org/10.7326/M18-0850>), quality assessment is considered optional in scoping reviews, depending on the aims of the review. Because our focus is on mapping the availability of evidence rather than quantifying it, we have opted not to undertake quality appraisal of individual studies. We believe this is in keeping with the best-practice guidance for scoping reviews" - in my opinion, this should be added in the manuscript so that the readers will get an idea of why the risk of bias has not been assessed.

The following was added under Data Charting:

Because our focus is on mapping the availability of evidence, we did not to undertake quality appraisal of individual studies.¹⁷

Reviewer: 2

The authors have attempted to revise the manuscript to the best of their abilities based on our earlier comments.

There are however, minor comments which needs to be addressed.

Abstract, line 3: the authors use " quality or financial protection" elsewhere the two are treated differently. Reading the manuscript, they are two different items so I suggest they replace "or" with "and" . This has been repeated on line 2 under " Eligibility"

Thank you for picking this up. In the introduction section of the abstract and where we state our aim in the Introduction section of the protocol we have used 'and' because we aim to map all four of these dimensions. We use 'or' in the methods section of the abstract and in the eligibility section of the protocol because we will include studies that report any one of these dimensions. i.e. the study does not have to include all 4 dimensions in order to be included. We believe use of the word 'or' here is the clearest way to convey this.

Data Charting and Extraction

The authors suggests that information that is absent or unclear will be addressed by contacting the study authors. It is however unclear what the authors will do to such data if they do not receive any feedback

The following was added to Data Charting:

The result of these attempts will be reported.